# Impact of Diabetes Mellitus on Cervical Spine Surgery for Ossification of the Posterior Longitudinal Ligament

**DOI:** 10.3390/jcm10153375

**Published:** 2021-07-29

**Authors:** Atsushi Kimura, Katsushi Takeshita, Toshitaka Yoshii, Satoru Egawa, Takashi Hirai, Kenichiro Sakai, Kazuo Kusano, Yukihiro Nakagawa, Kanichiro Wada, Keiichi Katsumi, Kengo Fujii, Takeo Furuya, Narihito Nagoshi, Tsukasa Kanchiku, Yukitaka Nagamoto, Yasushi Oshima, Hiroaki Nakashima, Kei Ando, Masahiko Takahata, Kanji Mori, Hideaki Nakajima, Kazuma Murata, Shunji Matsunaga, Takashi Kaito, Kei Yamada, Sho Kobayashi, Satoshi Kato, Tetsuro Ohba, Satoshi Inami, Shunsuke Fujibayashi, Hiroyuki Katoh, Haruo Kanno, Kota Watanabe, Shiro Imagama, Masao Koda, Yoshiharu Kawaguchi, Masaya Nakamura, Morio Matsumoto, Masashi Yamazaki, Atsushi Okawa

**Affiliations:** 1Department of Orthopedics, Jichi Medical University, 3311-1 Yakushiji, Shimotsuke 329-0498, Tochigi, Japan; dtstake@gmail.com; 2Japanese Multicenter Research Organization for Ossification of the Spinal Ligament, Tokyo 113-8510, Japan; yoshii.orth@tmd.ac.jp (T.Y.); egawa.orth@tmd.ac.jp (S.E.); hirai.orth@tmd.ac.jp (T.H.); kenitiro1122@gmail.com (K.S.); kz_kusano@yahoo.co.jp (K.K.); yukihiro19670916@gmail.com (Y.N.); wadak39@hirosaki-u.ac.jp (K.W.); kkatsu_os@yahoo.co.jp (K.K.); kengox15feb@gmail.com (K.F.); furuya-takeo@chiba-u.jp (T.F.); t-kanchi@pg8.so-net.ne.jp (T.K.); 7gam0to@gmail.com (Y.N.); yoo-tky@umin.ac.jp (Y.O.); hirospine@med.nagoya-u.ac.jp (H.N.); andokei@med.nagoya-u.ac.jp (K.A.); takamasa@med.hokudai.ac.jp (M.T.); kanchi@belle.shiga-med.ac.jp (K.M.); nhideaki@u-fukui.ac.jp (H.N.); kaz.mur26@gmail.com (K.M.); shunji622@gmail.com (S.M.); takashikaito@gmail.com (T.K.); yamada_kei@kurume-u.ac.jp (K.Y.); dr.shokobayashi@gmail.com (S.K.); skato323@gmail.com (S.K.); tooba@yamanashi.ac.jp (T.O.); iinami@dokkyomed.ac.jp (S.I.); shfuji@kuhp.kyoto-u.ac.jp (S.F.); hero@tokai-u.jp (H.K.); kanno-h@med.tohoku.ac.jp (H.K.); watakota@gmail.com (K.W.); imagama@med.nagoya-u.ac.jp (S.I.); masaokod@gmail.com (M.K.); zenji@med.u-toyama.ac.jp (Y.K.); masa@keio.jp (M.N.); morio@a5.keio.jp (M.M.); masashiy@md.tsukuba.ac.jp (M.Y.); okawa.orth@tmd.ac.jp (A.O.); 3Department of Orthopedic Surgery, Tokyo Medical and Dental University, 1-5-45 Yushima, Bunkyo Ward, Tokyo 113-8519, Japan; 4Department of Orthopedic Surgery, Saiseikai Kawaguchi General Hospital, 5-11-5 Nishikawaguchi, Kawaguchishi, Saitama 332-8558, Japan; 5Department of Orthopedic Surgery, Kudanzaka Hospital, 1-6-12 Kudanminami, Chiyodaku 102-0074, Japan; 6Department of Orthopaedic Surgery, Wakayama Medical University Kihoku Hospital, 219 Myoji, Katsuragi-cho, Itogun, Wakayama 649-7113, Japan; 7Department of Orthopedic Surgery, Hirosaki University Graduate School of Medicine, 5 Zaifucho, Hirosaki, Aomori 036-8562, Japan; 8Department of Orthopedic Surgery, Niigata University Medical and Dental General Hospital, 1-754 Asahimachidori, Chuo Ward, Niigata, Niigata 951-8520, Japan; 9Department of Orthopedic Surgery, Faculty of Medicine, University of Tsukuba, 1-1-1 Tennodai, Tsukuba, Ibaraki 305-8575, Japan; 10Department of Orthopedic Surgery, Chiba University Graduate School of Medicine, 1-8-1 Inohana, Chuo Ward, Chiba, Chiba 260-8670, Japan; 11Department of Orthopaedic Surgery, School of Medicine, Keio University, 35 Shinanomachi, Shinjuku Ward, Tokyo 160-8582, Japan; nagoshi@keio.jp; 12Department of Orthopedic Surgery, Yamaguchi University School of Medicine, 1144 Kogushi, Ube, Yamaguchi 755-8505, Japan; 13Department of Orthopedic Surgery, Osaka Rosai Hospital, 1179-3 Nagasonecho, Sakaishi, Osaka 591-8025, Japan; 14Department of Orthopaedic Surgery, Faculty of Medicine, The University of Tokyo, 7-3-1 Hongo, Bunkyo-ku, Tokyo 113-0033, Japan; 15Department of Orthopedic Surgery, Nagoya University Graduate School of Medicine, 65 Tsurumaicho, Showa Ward, Nagoya, Aichi 466-8550, Japan; 16Department of Orthopaedic Surgery, Faculty of Medicine and Graduate School of Medicine, Hokkaido University, Kita 15, Nishi 7, Sapporo 060-8638, Japan; 17Department of Orthopaedic Surgery, Shiga University of Medical Science, Tsukinowa-cho, Seta, Otsu, Shiga 520-2192, Japan; 18Department of Orthopaedics and Rehabilitation Medicine, Faculty of Medical Sciences University of Fukui, 23-3 Matsuoka Shimoaizuki, Eiheiji-cho, Yoshida-gun, Fukui 910-1193, Japan; 19Department of Orthopedic Surgery, Tokyo Medical University, 6-7-1 Nishishinjuku, Shinjuku-ku, Tokyo 160-0023, Japan; 20Department of Orthopedic Surgery, Imakiire General Hospital, 4-16 Shimotatsuocho, Kagoshimashi 892-8502, Japan; 21Department of Orthopedic Surgery, Graduate School of Medicine, Osaka University, 2-2 Yamadaoka, Suita-shi, Osaka 565-0871, Japan; 22Department of Orthopaedic Surgery, Kurume University School of Medicine, 67 Asahi-machi, Kurume-shi, Fukuoka 830-0011, Japan; 23Department of Orthopedic Surgery, Hamamatsu University School of Medicine, 1-20-1 Handayama, Hamamatsu, Shizuoka 431-3125, Japan; 24Department of Orthopaedic Surgery, Graduate School of Medical Sciences, Kanazawa University, 13-1 Takara-machi, Kanazawa 920-8641, Japan; 25Department of Orthopedic Surgery, University of Yamanashi, 1110 Shimokato, Chuo Ward, Yamanashi 409-3898, Japan; 26Department of Orthopaedic Surgery, Dokkyo Medical University School of Medicine, 880 Kitakobayashi, Mibu-machi, Shimotsuga-gun, Tochigi 321-0293, Japan; 27Department of Orthopaedic Surgery, Graduate School of Medicine, Kyoto University, 54 Kawahara-cho, Shogoin, Sakyo-ku, Kyoto 606-8507, Japan; 28Department of Orthopedic Surgery, Surgical Science, Tokai University School of Medicine, 143 Shimokasuya, Isehara, Kanagawa 259-1193, Japan; 29Department of Orthopaedic Surgery, Tohoku University School of Medicine, 1-1 Seiryomachi, Aoba Ward, Sendai, Miyagi 980-8574, Japan; 30Department of Orthopedic Surgery, Faculty of Medicine, University of Toyama, 2630 Sugitani, Toyama 930-0194, Japan

**Keywords:** ossification of the posterior longitudinal ligament, cervical myelopathy, diabetes mellitus, surgical outcome

## Abstract

Ossification of the posterior longitudinal ligament (OPLL) is commonly associated with diabetes mellitus (DM); however, the impact of DM on cervical spine surgery for OPLL remains unclear. This study was performed to evaluate the influence of diabetes DM on the outcomes following cervical spine surgery for OPLL. In total, 478 patients with cervical OPLL who underwent surgical treatment were prospectively recruited from April 2015 to July 2017. Functional measurements were conducted at baseline and at 6 months, 1 year, and 2 years after surgery using JOA and JOACMEQ scores. The incidence of postoperative complications was categorized into early (≤30 days) and late (>30 days), depending on the time from surgery. From the initial group of 478 patients, 402 completed the 2-year follow-up and were included in the analysis. Of the 402 patients, 127 (32%) had DM as a comorbid disease. The overall incidence of postoperative complications was significantly higher in patients with DM than in patients without DM in both the early and late postoperative periods. The patients with DM had a significantly lower JOA score and JOACMEQ scores in the domains of lower extremity function and quality of life than those without DM at the 2-year follow-up.

## 1. Introduction

Ossification of the posterior longitudinal ligament (OPLL) is a multifactorial disease that develops under complex genetic and environmental conditions [1,2]. The ectopic ossification causes chronic compression of the spinal cord, which leads to neurological dysfunction below the level of compression [3]. Although the prevalence of OPLL in the general population is relatively low, ranging from 0.1 to 2.5% in the United States and 1.9 to 4.3% in Japan [1,4,5], OPLL accounts for 18 to 35% as an etiology of degenerative cervical myelopathy (DCM), which requires surgical treatment [6]. Thus, OPLL is a major etiology of DCM, irrespective of race or region [6,7].

OPLL is associated with an increased prevalence of diabetes mellitus (DM) [5,8,9]. The prevalence of DM in patients with OPLL is 27% in the United States [5]. DM increases the prevalence of comorbidities, such as obesity, hypertension, common infections, and systemic vascular diseases [10], thereby exerting a negative impact on spinal surgeries [11]. Several studies have investigated the impact of DM on surgical outcomes in patients undergoing surgical treatment for cervical spondylotic myelopathy (CSM) or DCM (CSM and OPLL) [12,13]. Despite the high prevalence of DM in patients with OPLL, only a few retrospective case series have analyzed DM as a prognostic factor for the surgical management of cervical OPLL [14].

Recently, we conducted a multi-institutional prospective study to determine the outcomes of cervical spine surgery in patients with OPLL [15]. This study is a post hoc analysis of prospectively collected data. Thus, this study aimed to clarify the impact of DM on surgical outcomes in patients with OPLL.

## 2. Materials and Methods

### 2.1. Study Population

A multi-institutional prospective study to explore surgical outcomes of cervical OPLL was conducted by the Japanese Multicenter Research Organization for Ossification of the Spinal Ligament with the assistance of the Japanese Ministry of Health, Labor, and Welfare and the Japanese Agency for Medical Research and Development. This study was approved by all 28 institutions affiliated with the Japanese Multicenter Research Organization for Ossification of the Spinal Ligament. A total of 478 patients with cervical OPLL who underwent surgical treatment were prospectively recruited between April 2015 and July 2017. The exclusion criteria were as follows: (1) a history of cervical spine surgery and (2) neurological disturbance owing to disk herniation, infection, trauma, or spondylosis.

### 2.2. Surgical Procedures

The surgical procedures in this study can be categorized into the following four groups: anterior decompression and fusion (ADF), laminoplasty (LP), posterior decompression and fusion (PDF), and combined anterior and posterior fusion. LP included the following two representative methods: open-door and double-door LPs. ADF typically involved single- or multi-level discectomies and/or corpectomies. Anterior decompression of the spinal cord was achieved by complete resection or meticulous thinning (floating method) of the OPLL, followed by autogenous bone grafting with titanium plate fixation. PDF involved posterior decompression by laminectomy or LP, followed by instrumented posterior fixation. Combined anterior and posterior fusion was performed as a combination of ADF and PDF.

### 2.3. Functional Measurements

Functional measurements were conducted at baseline and at 6 months, 1 year, and 2 years after surgery using the Japanese Orthopedic Association (JOA) score and JOA Cervical Myelopathy Evaluation Questionnaire (JOACMEQ). The JOA score (17–2 version) evaluates the following seven categories of function: motor function of the fingers, shoulder and elbow, and lower extremity; sensory function of the upper extremity, trunk, and lower extremity; and function of the bladder [16]. Each category of function is scaled from 0 to 4, −2, 4, 2, 2, 2, and 3, respectively, with the total score ranging from −2 to 17. The JOACMEQ is a disease-specific patient-reported outcome measure for cervical myelopathy [17]. It is a self-administered questionnaire comprising 24 items, which evaluate the following 5 functional domains: (1) cervical spine function, (2) upper extremity function, (3) lower extremity function, (4) bladder function, and (5) quality of life (QOL), with each domain ranging from 0 to 100. Good reliability, validity, and responsiveness have been demonstrated in the original and translated versions of the JOACMEQ [18].

### 2.4. Postoperative Complications

Postoperative complications were defined as any unexpected or undesirable events occurring as a direct or indirect result of surgery. All complications were prospectively identified during the 2-year follow-up period. The incidence of postoperative complications was categorized into early (≤30 days) and late (>30 days) depending on the time from surgery.

### 2.5. Imaging Studies

Preoperative radiological examinations were performed using plain X-ray, computed tomography, and magnetic resonance imaging in all patients. The Cobb angle between C2 and C7 (C2–C7 Cobb angle), C2–C7 range of motion, occupying rate of OPLL, and K-line (+/−) were evaluated using preoperative plain X-ray images [19].

### 2.6. Preoperative Glycemic Control and Treatment Modalities of DM

Preoperative treatment modalities for DM were investigated using the patients’ medical records. The treatment modalities were categorized into the following three groups: dietary control, oral antidiabetics, and insulin treatment. As an indicator of preoperative glycemic control in patients with DM, we collected data on the preoperative level of glycated hemoglobin A1c (HbA1c), which reflects the average blood glucose level during the previous 2 to 3 months.

### 2.7. Statistical Analysis

All statistical analyses were performed using Statistical Package for the Social Sciences, version 22 (IBM Corp., Armonk, NY, USA). The *p* values were calculated using the unpaired *t*-test or the Mann–Whitney U test for means, Fisher’s exact test for proportions, and the Wilcoxon signed-rank test for medians. The Shapiro–Wilk test for normality was used to choose between the unpaired *t*-test and the Mann–Whitney U test. Multiple comparisons were performed using one-way analysis of variance (ANOVA) with Tukey’s post hoc test. Values of *p* < 0.05 were considered to indicate a statistically significant difference.

## 3. Results

### 3.1. Patients’ Characteristics and Baseline Functions

From the initial group of 478 participants, 402 completed the 2-year follow-up (follow-up rate: 84%) and were included in the analysis. Of the 402 patients, 127 (32%) had DM as a comorbid disease. The 127 patients with DM comprised 123 patients with non-insulin-dependent DM and 4 patients with insulin-dependent DM. The comparisons of patients’ demographics and baseline functions between patients with and without DM are summarized in Table 1. The patients with DM had a significantly higher body mass index (BMI) and rates of hypertension, myocardial infarction, and anticoagulant/antiplatelet medication than those without DM. Additionally, the patients with DM had significantly inferior lower extremity function measured using the JOACMEQ and significantly higher visual analog scale scores for neck pain than those without DM. Regarding surgical methods, the patients with DM had a significantly higher rate of PDF and a significantly higher number of surgical levels than those without DM.

### 3.2. Postoperative Complications

The incidence of postoperative complications is shown in Table 2. Each early complication showed no significant difference between the groups, except urinary tract infection; however, the overall incidence of early complications was significantly higher in the patients with DM than in the patients without DM. Similarly, the overall incidence of late complications was significantly higher in the patients with DM than in the patients without DM.

### 3.3. Functional Outcomes

The functional outcomes at the 2-year follow-up are presented in Table 3. The patients with DM had a significantly lower JOA score than those without DM. Furthermore, the patients with DM had significantly lower JOACMEQ scores in the domains of lower extremity function and QOL than the patients with DM. The comparisons of postoperative functional gain between the patients with and without DM are summarized in Table 4. The average postoperative gains of functional scores were consistently lower in the patients with DM than in those without DM; however, the difference did not reach statistical significance in any functional measures.

### 3.4. Time-Dependent Change of Functional Outcomes

The time-dependent change in the JOA score is shown in Figure 1. The average JOA score was consistently lower in the patients with DM than in the patients without DM during the observation period. The difference between the groups increased over time and reached statistical significance at the 2-year follow-up. Similarly, the average scores of the five JOACMEQ domains were lower in the patients with DM than in the patients without DM (Figure 2). The difference in lower extremity function was significant during the observation period, except for the 1-year follow-up (*p* = 0.056). The difference in upper extremity function showed statistical significance at the 6-month follow-up and returned to a comparable level at the 1-year follow-up or later. Similar to the change in the JOA score, the difference in the QOL domain of the JOACMEQ between the groups increased over time and reached statistical significance at the 2-year follow-up.

### 3.5. Surgical Outcomes Stratified by Surgical Procedures

Surgical outcomes were compared among the four surgical procedures (Table 5). The PDF group had a significantly lower preoperative JOA score than the LP group (*p* = 0.007, one-way ANOVA followed by Tukey’s post hoc test). The ADF group showed a significantly higher postoperative JOA score than the PDF group (*p* = 0.011, one-way ANOVA followed by Tukey’s post hoc test). However, the recovery rate of the JOA score showed no significant difference among the groups. The incidence of early postoperative complications was significantly higher in the ADF and PDF groups than in the LP group.

### 3.6. Surgical Outcomes Stratified by Treatment Modalities for DM

Surgical outcomes were stratified into three groups based on the treatment modalities (Table 6). The preoperative HbA1c level differed significantly among the groups. The patients in the insulin therapy group had a significantly higher preoperative HbA1c level than the patients in the other groups (*p* < 0.001, one-way ANOVA followed by Tukey’s post hoc test). However, neither functional outcomes nor the incidence of postoperative complications showed significant differences among the groups.

## 4. Discussion

In the present study, we investigated the influence of DM on surgical outcomes in patients undergoing surgical treatment for cervical OPLL. The key findings of this study are as follows. (1) Of the 402 patients who underwent surgical treatment for cervical OPLL, 127 (32%) had DM as a comorbid disease. (2) The patients with DM had a significantly higher BMI and comorbid rates of hypertension, myocardial infarction, and anticoagulant/antiplatelet medication than those without DM. (3) The patients with DM had a significantly higher rate of complications than those without DM in both the early (≤30 days) and late (>30 days) postoperative periods. (4) The patients with DM had a significantly lower JOA score and JOACMEQ scores in the domains of lower extremity function and QOL than those without DM at the 2-year follow-up.

In recent decades, the prevalence of DM has increased substantially worldwide, and population aging is an important factor influencing this increasing trend. The prevalence of DM in the Japanese population aged 60 to 69 years in 2015 was 18 and 10% in men and women, respectively [20]. The 32% prevalence of DM in this cohort was approximately twice the average prevalence of DM in the Japanese population. Consistent with our findings, Bakhsh et al. [5] demonstrated that the prevalence of DM differs significantly between patients with and without OPLL (27.0% vs. 13.0%, respectively) in a large-scale retrospective cohort in the United States. These results suggest that patients with OPLL are more likely to be affected by DM than are patients with other forms of cervical myelopathy.

DM substantially increases the risk of developing systemic complications after surgical interventions through several pathological mechanisms. First, DM impairs the function of endothelial and vascular smooth muscle cells, which leads to systemic atherosclerosis and its complications, such as cardiac infarction, cerebral infarction, peripheral vascular diseases, nephropathy, and retinopathy [10]. Second, type 2 DM is associated with increased comorbid rates of obesity and hypertension, and the coexistence of these conditions further increases the risk of macrovascular and microvascular complications [21]. Finally, DM increases patients’ susceptibility to infections, such as lower respiratory tract infection, urinary tract infection, and skin and mucous membrane infection, because of the impaired innate and adaptive immune responses against invading pathogens [22]. Indeed, in this study, the patients with DM showed a significantly higher BMI and rate of hypertension, myocardial infarction, and anticoagulant/antiplatelet medication than those without DM at baseline. Furthermore, the patients with DM had a significantly higher incidence of urinary tract infection in the early postoperative period. These results indicate that patients with OPLL have an increased risk of systemic complications after surgery. Although we could not find statistical significance, except for urinary tract infection, the significantly higher overall incidence of complications may be attributed to various systemic comorbidities associated with DM.

The impact of DM on surgical outcomes after spine surgery remains controversial. Armaghani et al. [23] demonstrated that DM is associated with worse patient-reported outcomes, such as the Neck Disability Index and EuroQOL-5 Dimensions, when patients with DM were compared with those without DM following elective cervical spine surgery. However, Arnold et al. [24] concluded that the outcomes of surgical decompression for CSM are similar in patients with and without DM, except for the 36-Item Short Form Health Survey Physical Functioning scores. More recently, Nori et al. [13] showed that patients with CSM who had DM experienced improvements in neurological function following posterior decompression to the same extent observed in those without DM. This study focused exclusively on patients with OPLL and showed comparable neurological improvements measured using both the JOA score and JOACMEQ scores between patients with and without DM, although the functional outcomes at the 2-year follow-up were significantly worse in patients with DM than in those without DM. The significantly worse functional outcomes at the endpoint were partly attributable to the significantly worse lower extremity function at baseline in patients with DM than in patients without DM. The significantly lower baseline physical function is consistent with that in past studies that demonstrate that older individuals with DM are associated with weaker muscle strength and a higher risk of impaired physical function than their age-matched counterparts without DM [25,26]. Furthermore, the significantly higher rate of medical comorbidities and the significantly higher incidence of postoperative complications in patients with DM may interfere with functional outcomes. Although most of the medical comorbidities and postoperative complications are not directly associated with neurological functions, poorer general health conditions might impair not only physical functions but also QOL. Indeed, patients with DM had a significantly lower score in the QOL domain of the JOACMEQ than patients without DM.

A recent systematic review demonstrated that HbA1c is predictive for postoperative infection and functional outcomes in patients undergoing spine surgery and that an HbA1c level of >6.5 to 6.9% is associated with an increased risk of postoperative complications. However, in the present study, the preoperative HbA1c level showed no significant association with either functional outcomes or the incidence of postoperative complications. One possible explanation for this inconsistency may be the small number of patients with DM in the present study. Our sample size may be insufficient to detect the predictive value of HbA1c on surgical outcomes. Consistent with our results, Nagoshi et al. [27] recently showed no significant correlation between the preoperative HbA1c level and postoperative JOA score in 47 patients with concurrent OPLL and DM. Furthermore, some patients with a high preoperative HbA1c level received perioperative insulin therapy. Rigorous glycemic control during the perioperative period might reduce the risk of postoperative complications associated with poorly controlled DM, resulting in a reduced predictive accuracy of the preoperative HbA1c level. A further large-scale prospective study is required to clarify the predictive value of HbA1c and its optimal cut-off point to identify patients with an increased risk of postoperative complications.

This study has several potential limitations. First, the significant difference in the surgical methods between the two groups may affect the rate of postoperative complications. Patients with DM had a higher rate of PDF and a lower rate of anterior decompression and fusion (ADF). As multilevel ADF, especially fusion for three or more levels, is associated with a higher incidence of perioperative complications than PDF [28,29], the difference in surgical methods may mask the risk of perioperative complications in patients with DM. Second, the relatively small sample size in this cohort may have led to a type II error in the analysis of perioperative complications. The incidence of individual complications was largely less than a few percentage points; the sample size in this study may be insufficient to detect the impact of DM on each complication. Likewise, the number of patients with DM may be insufficient to show the usefulness of HbA1c levels for identifying an increased risk of postoperative complications [11,30,31].

## 5. Conclusions

About one-third of the patients with OPLL undergoing surgery had comorbid DM, which was associated with a significantly higher overall incidence of postoperative complications and significantly worse functional outcomes at the 2-year follow-up than the absence of DM. As preoperative glycemic control is considered a modifiable risk factor for postoperative complications [32], strict glycemic control along with careful preoperative medical checkups may minimize the risk of postoperative complications in patients with concurrent OPLL and DM.

## Figures and Tables

**Figure 1 jcm-10-03375-f001:**
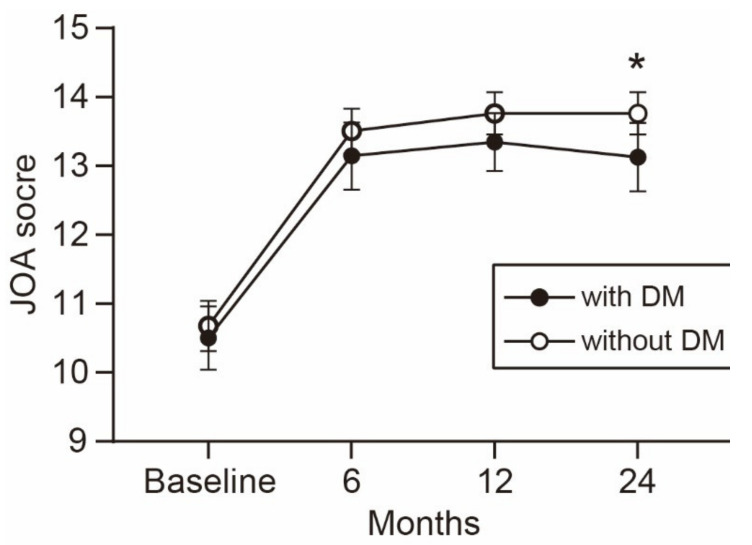
Time-dependent change in the JOA score. The average JOA score was consistently lower in patients with than without DM during the observation period. The difference between the groups increased over time and reached statistical significance at the 2-year follow-up. * *p* < 0.05, unpaired *t*-test. JOA, Japanese Orthopedic Association; DM, diabetes mellitus.

**Figure 2 jcm-10-03375-f002:**
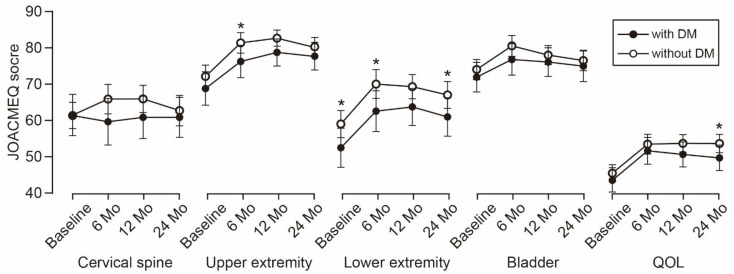
Time-dependent change in JOACMEQ scores. The difference in lower extremity function was significant during the observation period, except for the 1-year follow-up. The difference in upper extremity function showed statistical significance at the 6-month follow-up. The difference in the QOL domain between the groups increased over time and reached statistical significance at the 2-year follow-up. * *p* < 0.05, unpaired *t*-test. JOACMEQ, Japanese Orthopedic Association Cervical Myelopathy Evaluation Questionnaire; DM, diabetes mellitus; QOL, quality of life.

**Table 1 jcm-10-03375-t001:** Patients’ characteristics and baseline functions.

Characteristics	With DM (N = 127)	Without DM (N = 275)	*p* Value *
Age	64.2 ± 11.0	64.0 ± 11.9	0.743
Gender (Male/Female)	90/37	208/67	0.328
BMI	26.5 ± 4.5	25.3 ± 4.3	0.006
Medical comorbidities			
Hypertension	59 (46)	94 (34)	0.021
Cerebral infarction	11 (9)	10 (4)	0.051
Myocardial infarction	8 (6)	6 (2)	0.044
Musculoskeletal disease	18(14)	31 (11)	0.416
Connective tissue disease	1 (0.8)	3 (1.1)	1.000
Anticoagulant/antiplatelet medication	26 (20)	34 (12)	0.049
Duration of symptoms (month)	46.7 ± 62.3	41.4 ± 66.0	0.189
JOA score	10.5 ± 2.6	10.7 ± 3.0	0.341
JOACMEQ			
Cervical spine	60.8 ± 31.2	61.5 ± 28.7	0.944
Upper extremity	68.8 ± 25.2	72.6± 24.3	0.126
Lower extremity	52.4 ± 29.4	59.4 ± 29.4	0.019
Bladder	71.5 ± 23.6	72.9 ± 22.2	0.430
QOL	42.5 ± 19.8	44.4 ± 18.3	0.179
Neck pain VAS	47.6 ± 32.3	39.8 ± 30.7	0.035
Imaging finding			
C2-C7 Cobb angle (degree)	8.6 ± 12.9	10.1 ± 10.5	0.258
Range of motion (degree)	24.7 ± 12.4	28.1 ± 14.4	0.051
Occupancy ratio of OPLL (%)	45.7 ± 15.6	43.3 ± 15.3	0.173
K-line (−)	42 (33)	93 (34)	0.883
Surgical method			
ADF	21 (17)	68 (25)	0.071
PDF	39 (31)	52 (19)	0.010
LP	63 (50)	148 (54)	0.453
APF	4 (3)	7 (3)	0.748
No. of surgical levels	4 (3–5)	4 (3–4)	0.029

Data are shown as mean ± standard deviation, number (%), or median (25–75th percentile). * *p* values were calculated using the unpaired *t*-test for means, Fisher’s exact test for proportions, or the Wilcoxon signed rank test for medians. DM, diabetes mellitus; BMI, body mass index; JOA, Japanese Orthopedic Association; JOACMEQ, Japanese Orthopedic Association Cervical Myelopathy Evaluation Questionnaire; QOL, quality of life; VAS, visual analog scale; OPLL, ossification of the posterior longitudinal ligament; ADF, anterior decompression and fusion; PDF, posterior decompression and fusion; LP, laminoplasty; APF, combined anterior and posterior fusion.

**Table 2 jcm-10-03375-t002:** Incidence of early and late postoperative complications.

Complication	With DM (N = 127)	Without DM (N = 275)	*p* Value *
Early (≤30 days from surgery)			
Neurological deterioration	12 (9)	30 (11)	0.727
CSF leakage	6 (5)	14 (5)	1.000
Dysphasia	3 (2)	8 (3)	1.000
Graft bone failure	3 (2)	5 (2)	0.712
Instrument failure	2 (1.6)	5 (2)	1.000
Wound infection	2 (1.6)	4 (1.5)	1.000
Wound dehiscence	2 (1.6)	2 (0.7)	0.594
Epidural hematoma	1 (0.8)	1 (0.4)	0.533
Upper air way obstruction	1 (0.8)	1 (0.4)	0.533
Urinary tract infection	6 (5)	3 (1)	0.031
Delirium	4 (3)	5 (2)	0.472
Deep vein thrombosis	1 (0.8)	2 (0.7)	1.000
Gastrointestinal bleeding	0 (0)	3 (1)	0.555
Heart failure	1 (0.8)	1 (0.4)	0.533
Liver dysfunction	1 (0.8)	1 (0.4)	0.533
Brain infarction	0 (0)	1 (0.4)	1.000
Pneumonia	0 (0)	1 (0.4)	1.000
Cholecystitis	1 (0.8)	0 (0)	0.316
Any early complications	43 (34)	66 (24)	0.041
Late (>30 days from surgery)			
Instrument failure	6 (5)	5 (2)	0.109
Lumbar spinal stenosis	3 (2)	8 (3)	1.000
Adjacent segment disease	2 (1.6)	1 (0.4)	0.236
Thoracic OPLL	1 (0.8)	1 (0.4)	0.533
C5 palsy	1 (0.8)	1 (0.4)	0.528
Non-union	0 (0)	2 (0.7)	1.000
Wound infection	0 (0)	2 (0.7)	1.000
Dysphasia	2 (1.6)	0 (0)	0.099
Stroke	2 (1.6)	0 (0)	0.099
Urinary tract infection	2 (1.6)	1 (0.4)	0.236
Pneumonia	2 (1.6)	0 (0)	0.099
Parkinson’s disease	0 (0)	2 (0.7)	1.000
Multiple sclerosis	0 (0)	1 (0.4)	1.000
Any late complications	19 (15)	21 (8)	0.031

Data are shown as number (%). * *p* values were calculated using Fisher’s exact test. DM, diabetes mellitus; CSF, cerebrospinal fluid; OPLL, ossification of the posterior longitudinal ligament; C5 palsy, 5th cervical spinal nerve palsy.

**Table 3 jcm-10-03375-t003:** Comparisons of functional outcomes between patients with and without diabetes mellitus at the 2-year follow-up.

Outcome	With DM (N = 127)	Without DM (N = 275)	*p* Value *
JOA score	13.1 ± 2.8	13.8 ± 2.5	0.024
JOACMEQ			
Cervical spine	60.7 ± 20.1	62.8 ± 32.1	0.389
Upper extremity	77.9 ± 20.0	81.1 ± 19.5	0.105
Lower extremity	61.0 ± 28.1	67.7 ± 28.1	0.026
Bladder	75.1 ± 22.4	77.1 ± 21.3	0.369
QOL	49.6 ± 18.7	54.1 ± 19.2	0.036
Neck pain VAS	40.6 ± 31.1	36.0 ± 30.0	0.197

Data are shown as mean ± standard deviation. * *p* values were calculated using the Mann–Whitney U test. DM, diabetes mellitus; JOA, Japanese Orthopedic Association; JOACMEQ, Japanese Orthopedic Association Cervical Myelopathy Evaluation Questionnaire; QOL, quality of life; VAS, visual analog scale.

**Table 4 jcm-10-03375-t004:** Comparisons of postoperative functional gain between patients with and without diabetes mellitus.

Outcome	With DM (N = 127)	Without DM (N = 275)	*p* Value *
Recovery rate of JOA score	40.8 ± 33.6	48.7 ± 32.5	0.051
Postoperative gain in JOA score	2.6 ± 2.4	3.1 ± 2.5	0.151
Postoperative gain in JOACMEQ			
Cervical spine	−3.1 ± 29.2	1.6 ± 34.1	0.230
Upper extremity	6.4 ± 23.1	8.4 ± 22.1	0.251
Lower extremity	6.2 ± 24.5	7.6 ± 24.2	0.826
Bladder	0.9 ± 20.7	3.0 ± 19.8	0.639
QOL	4.9 ± 16.7	9.3 ± 18.9	0.094
Postoperative change in neck pain VAS	−5.8 ± 32.6	−3.2 ± 33.7	0.625

Data are shown as mean ± standard deviation. * *p* values were calculated using the Mann–Whitney U test. DM, diabetes mellitus; JOA, Japanese Orthopedic Association; JOACMEQ, Japanese Orthopedic Association Cervical Myelopathy Evaluation Questionnaire; QOL, quality of life; VAS, visual analog scale.

**Table 5 jcm-10-03375-t005:** Comparisons of surgical outcomes stratified by surgical procedures.

	Surgical Procedure				*p* Value
	ADF (N = 89)	LP (N = 211)	PDF (N = 91)	APF (N = 11)	
No. of levels decompressed	3	4	5	4	<0.001 *
No. of levels fused	3	N/A	5	4	<0.001 *
Preoperative JOA score	10.9 ± 2.6	10.9 ± 2.7	9.7 ± 3.3	9.5 ± 3.2	0.002 *
Postoperative JOA score	14.1 ± 2.4	13.7 ± 2.4	12.9 ± 3.1	12.7 ± 2.7	0.010 *
Recovery rate of JOA score	53.1 ± 31.1	44.3 ± 33.7	44.9 ± 33.1	38.7 ± 30.4	0.157 *
Early complication	34 (38)	38 (18)	34 (37)	3 (27)	<0.001 ^†^
Late complication	12 (13)	15 (7)	13 (14)	0 (0)	0.067 ^†^

Data are shown as median, mean ± standard deviation, or number (%). * One-way analysis of variance. ^†^ Chi-square test. ADF, anterior decompression and fusion; LP, laminoplasty; PDF, posterior decom-pression and fusion; APF, combined anterior and posterior fusion; JOA, Japanese Orthopedic Association.

**Table 6 jcm-10-03375-t006:** Comparisons of surgical outcomes stratified by treatment modalities for diabetes mellitus.

	Treatment Modality			*p* Value
	Dietary Control (N = 32)	Oral Antidiabetics (N = 78)	Insulin Therapy (N = 17)	
Preoperative HbA1c	6.7 ± 1.3	6.7 ± 0.8	7.7 ± 1.1	<0.001 *
Preoperative JOA score	10.7 ± 3.3	10.5 ± 2.2	10.2 ± 3.1	0.788 *
Postoperative JOA score	13.3 ± 3.5	13.1 ± 2.6	13.2 ± 2.7	0.917 *
Recovery rate of JOA score	42.6 ± 29.0	38.6 ± 36.9	47.6 ± 25.3	0.576 *
Early (≤30 days from surgery) complications	10 (31)	29 (38)	5 (24)	0.518 ^†^
Late (>30 days from surgery) complication	2 (6)	16 (21)	1 (6)	0.086 ^†^

Data are shown as mean ± standard deviation or number (%). * One-way analysis of variance. ^†^ Chi-square test. HbA1c, glycated hemoglobin; JOA, Japanese Orthopedic Association.

## Data Availability

The data presented in this study are available on request from the corresponding author. The data are not publicly available due to privacy and ethical concerns.

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
