# Peer review of "Impact of Diabetes Mellitus on Cervical Spine Surgery for Ossification of the Posterior Longitudinal Ligament"

_jcm, 2021, doi:10.3390/jcm10153375_

Round 1
Reviewer 1 Report
This is an interesting retrospective study investigating the effect of diabetes mellitus on outcomes and complications following cervical spine surgery in patients affected by OPLL. The study is well written and structured. The methodology is scientifically sound and the results support the conclusions.
Some comments:
- A paragraph on the surgical techniques performed should be added with different approaches explained in detail (APF, ADF, PDF, laminoplasty). Furthermore, results should be stratified based upon the technique itself. Although mentioned in the limitations section, functional outcomes and post-operative complications may be significantly influenced in either group by the use of instrumentation and the extent of decompression. Not specifying these data can introduce a significant bias to the results.
- English language editing with a native speaker is strongly advised. Some lexical choices are inappropriate and some passages seem difficult to understand (e.g. line 99, line 102),
- Table 1 and 2: Please refer to "cerebral infarction" as "stroke".
- Table 2: What do authors mean for "neurodegenerative disorder"?
Author Response
Point 1: A paragraph on the surgical techniques performed should be added with different approaches explained in detail (APF, ADF, PDF, laminoplasty). Furthermore, results should be stratified based upon the technique itself. Although mentioned in the limitations section, functional outcomes and post-operative complications may be significantly influenced in either group by the use of instrumentation and the extent of decompression. Not specifying these data can introduce a significant bias to the results.
Response 1: We have added a paragraph that explains the details of each surgical procedure (page 3 lines 130–139). We also stratified the patients into four groups based on the surgical procedures performed and compared functional outcomes and postoperative complications among the groups. We found significant differences in the preoperative JOA score, postoperative JOA score, and incidence of early postoperative complications (Table 5). These results have been incorporated into the revised manuscript (page 5, lines 227–234).
Point 2: English language editing with a native speaker is strongly advised. Some lexical choices are inappropriate and some passages seem difficult to understand (e.g. line 99, line 102).
Response 2: We have had the manuscript revised by a native English speaker. All language modifications are marked in the revised manuscript using the track changes function.
Point 3: Table 1 and 2: Please refer to "cerebral infarction" as "stroke". Table 2: What do authors mean for "neurodegenerative disorder"?
Response 3: We have changed “cerebral infarction” to “stroke.” The neurodegenerative disorders included Parkinson’s disease in two patients and multiple sclerosis in one patient. We used specific disease names in the revised Table 1.

Reviewer 2 Report
Authors present a multiinstitutional prospective study on 402 cervical OPLL patients who underwent surgical treatment with functional measurments at baseline, 6 months, 1 year, and 2 years after surgery using JOA and JOACMEQ scores. The incidence of postoperative complications was categorized into early (≤30 82days) and late (>30 days) depending on the time from surgery. Out of this number, 127 patients had DM. The overall incidence of postoperative complications was found to be significantly higher in patients with DM than that in patients without DM in both early and late postoperative periods. Patients with DM had a significantly lower JOA score and JOACMEQ scores in the domains of lower extremity function and quality of life than those without DM at the 2-year follow.
Authors present an interesting aspect of treatment of OPLL with diabetes mellitus being identified as a signficant risk factor which increases the complications.
Several issues need to be adressed:
How was the status of the patients with DM evaluated? It would be useful to include the status of the disease for all 127 patients using Hb1ac and the treatment modality (insulin or oral antidiabetics) or other method in order to atain the severity of the disease to the complication rate. You have included this in the limitations section; however a certain stratification of the severity of the disease is needed.
In your opinion, is DM worsening the OPLL as a disease itself or surgical treatment of OPLL is being affected by DM in terms of increased complication rate?
I recommend including following articles in for the discussion:
-Nagoshi N, Watanabe K, Nakamura M, Matsumoto M, Li N, Ma S, He D, Tian W, Jeon H, Lee JJ, Kim KN, Ha Y, Hong Kwan KY, Po Cheung AK. Does Diabetes Affect the Surgical Outcomes in Cases With Cervical Ossification of the Posterior Longitudinal Ligament? A Multicenter Study From Asia Pacific Spine Study Group. Global Spine J. 2021 Mar 10:2192568221996300. doi: 10.1177/2192568221996300. Epub ahead of print. PMID: 33715508.
-Kobashi G, Washio M, Okamoto K, Sasaki S, Yokoyama T, Miyake Y, Sakamoto N, Ohta K, Inaba Y, Tanaka H; Japan Collaborative Epidemiological Study Group for Evaluation of Ossification of the Posterior Longitudinal Ligament of the Spine Risk. High body mass index after age 20 and diabetes mellitus are independent risk factors for ossification of the posterior longitudinal ligament of the spine in Japanese subjects: a case-control study in multiple hospitals. Spine (Phila Pa 1976). 2004 May 1;29(9):1006-10. doi: 10.1097/00007632-200405010-00011. PMID: 15105673. Important limitations of the study are noted in the discussion section, however this study sheds a new light on a thin body of literature concerning connection between OPLL and DM. It would be useful to add more sources to the discussion which evaluate influence of DM on patient outcomes in spine surgery, especially in cervical spine surgery. I recommend to include following study for citation: Mullins J, Pojskić M, Boop FA, Arnautović KI. Retrospective single-surgeon study of 1123 consecutive cases of anterior cervical discectomy and fusion: a comparison of clinical outcome parameters, complication rates, and costs between outpatient and inpatient surgery groups, with a literature review. J Neurosurg Spine. 2018 Jun;28(6):630-641. doi: 10.3171/2017.10.SPINE17938. Epub 2018 Mar 30. PMID: 29600910.Author Response
Point 1: How was the status of the patients with DM evaluated? It would be useful to include the status of the disease for all 127 patients using Hba1c and the treatment modality (insulin or oral antidiabetics) or other method in order to attain the severity of the disease to the complication rate. You have included this in the limitations section; however, a certain stratification of the severity of the disease is needed.
Response 1: We agree with the reviewer that stratifying the severity of diabetes would provide important information for analysis of the complication rate. We collected additional data regarding the preoperative HbA1c levels and treatment modalities for DM (page 4, lines 167–173). The treatment modalities were stratified into three groups: dietary control, oral antidiabetics, and insulin treatment (Table 6). Patients in the insulin therapy group had a significantly higher preoperative HbA1c level than patients in the other groups. However, neither functional outcomes nor the incidence of postoperative complications showed a significant difference among the groups. These results have been incorporated into the revised manuscript (page 5, lines 236–242; page 12, lines 343–358; page 12, lines 368–370).
Point 2: In your opinion, is DM worsening the OPLL as a disease itself or surgical treatment of OPLL is being affected by DM in terms of increased complication rate?
Response 2: Given the comparable occupancy ratio of OPLL between patients with and without DM, we speculate that DM exerts a negative effect on surgical outcomes largely through indirect mechanisms, such as increased medical comorbidities, lower baseline physical functions, and a higher incidence of postoperative complications, rather than directly enhancing the progression of OPLL lesions. We have modified the discussion regarding the impact of DM on functional outcomes (page 11, lines 334–340).
Point 3: I recommend including following articles in for the discussion:
-Nagoshi N, Watanabe K, Nakamura M, Matsumoto M, Li N, Ma S, He D, Tian W, Jeon H, Lee JJ, Kim KN, Ha Y, Hong Kwan KY, Po Cheung AK. Does Diabetes Affect the Surgical Outcomes in Cases With Cervical Ossification of the Posterior Longitudinal Ligament? A Multicenter Study From Asia Pacific Spine Study Group. Global Spine J. 2021 Mar 10:2192568221996300. doi: 10.1177/2192568221996300. Epub ahead of print. PMID: 33715508.
-Kobashi G, Washio M, Okamoto K, Sasaki S, Yokoyama T, Miyake Y, Sakamoto N, Ohta K, Inaba Y, Tanaka H; Japan Collaborative Epidemiological Study Group for Evaluation of Ossification of the Posterior Longitudinal Ligament of the Spine Risk. High body mass index after age 20 and diabetes mellitus are independent risk factors for ossification of the posterior longitudinal ligament of the spine in Japanese subjects: a case-control study in multiple hospitals. Spine (Phila Pa 1976). 2004 May 1;29(9):1006-10. doi: 10.1097/00007632-200405010-00011. PMID: 15105673.
Response 3: We have cited the above-mentioned articles and have added relevant text to the Discussion section (page 3, lines 104–105; page 12, lines 350–352).
Point 4: Important limitations of the study are noted in the discussion section, however this study sheds a new light on a thin body of literature concerning connection between OPLL and DM. It would be useful to add more sources to the discussion which evaluate influence of DM on patient outcomes in spine surgery, especially in cervical spine surgery. I recommend to include following study for citation:
Mullins J, Pojskić M, Boop FA, Arnautović KI. Retrospective single-surgeon study of 1123 consecutive cases of anterior cervical discectomy and fusion: a comparison of clinical outcome parameters, complication rates, and costs between outpatient and inpatient surgery groups, with a literature review. J Neurosurg Spine. 2018 Jun;28(6):630-641. doi: 10.3171/2017.10.SPINE17938. Epub 2018 Mar 30. PMID: 29600910.
Response 4: We have cited the above-mentioned article in the Discussion section (page 12, lines 362–363).

Round 2
Reviewer 2 Report
The authors have sufficiently responded to the requests and made significant changes to the manuscript. English language editing or check of the text with native speaker makes it suitable for publication.